

# Analysis of genes and underlying mechanisms involved in foam cells formation and atherosclerosis development

Kai Zhang, Xianyu Qin, Xianwu Zhou, Jianrong Zhou, Pengju Wen, Shaoxian Chen, Min Wu, Yueheng Wu and Jian Zhuang

Department of Cardiovascular Surgery, Guangdong Provincial Key Laboratory of South China Structural Heart Disease, Guangdong Provincial People's Hospital & Guangdong Academy of Medical Sciences, School of Medicine, South China University of Technology, Guangdong Cardiovascular Institute, Guangzhou, Guangdong, China

## ABSTRACT

**Background**. Foam cells (FCs) play crucial roles in the process of all stages of atherosclerosis. Smooth muscle cells (SMCs) and macrophages are the major sources of FCs. This study aimed to identify the common molecular mechanism in these two types of FCs.

**Methods**. GSE28829, GSE43292, GSE68021, and GSE54666 were included to identify the differentially expressed genes (DEGs) associated with FCs derived from SMCs and macrophages. Gene Ontology biological process (GO-BP) and Kyoto Encyclopedia of Genes and Genomes (KEGG) pathway analyses were performed by using the DAVID database. The co-regulated genes associated with the two origins of FCs were validated (GSE9874), and their expression in vulnerable atherosclerosis plaques (GSE120521 and GSE41571) was assessed.

**Results**. A total of 432 genes associated with FCs derived from SMCs (SMC-FCs) and 81 genes associated with FCs derived from macrophages (M-FCs) were identified, and they were mainly involved in lipid metabolism, inflammation, cell cycle/apoptosis. Furthermore, three co-regulated genes associated with FCs were identified: *GLRX*, *RNF13*, and *ABCA1*. These three common genes showed an increased tendency in unstable or ruptured plaques, although in some cases, no statistically significant difference was found.

**Conclusions**. DEGs related to FCs derived from SMCs and macrophages have contributed to the understanding of the molecular mechanism underlying the formation of FCs and atherosclerosis. *GLRX*, *RNF13*, and *ABCA1* might be potential targets for atherosclerosis treatment.

# INTRODUCTION

Atherosclerosis is a complex chronic disease characterized by the thickening of the intima and the formation of atherosclerotic plaques, leading to asymmetric stenosis of the

Corresponding authors
Yueheng Wu,
edgar_wuyh@outlook.com
Jian Zhuang,
zhuangjiangenetics@163.com

arterial lumen (*Wang et al., 2019a*). It is the underlying pathophysiological mechanism of many cardiac-cerebrovascular diseases, including coronary arterial disease, ischemic stroke, and other peripheral arterial diseases, which are the leading causes of death in the world (*Katakami, 2018*; *Wang et al., 2019a*). The pathogenesis and molecular mechanisms involved in atherosclerosis are multifactorial, including endothelial dysfunction, abnormal smooth muscle cell (SMC) proliferation and migration, oxidized lipid deposition, vascular matrix changes, inflammatory cell infiltration, and oxidative stress (*Weber & Noels, 2011*; *Zhou et al., 2019*). In recent decades, significant improvements have been made in the diagnosis, prevention, and treatment of atherosclerosis. However, the comprehensive and in-depth molecular mechanisms of atherosclerosis still need to be further explored (*Kanter et al., 2018*).

Intra- and extracellular lipid deposition is a crucial trigger for the development of atherosclerotic lesions (*Summerhill et al., 2019*), while lipid-rich foam cells (FCs) are the earliest manifestations of atherosclerotic lesions, and their emergence and accumulation are the original causes of atherosclerotic plaques (*Olivares, González Ballester & Noailly, 2016*). It was previously thought that all FCs in human atherosclerotic lesions were derived from macrophages. However, subsequent studies of human arterial plaques showed that cell types inherent to the arterial wall are also involved in the formation of FCs, including SMCs, stem/progenitor cells, and even endothelial cells (*Gisterå & Hansson, 2017*; *Maguire, Pearce & Xiao, 2019*). SMCs and macrophages are the two main sources of FCs in arterial plaques. FCs formation in atherosclerosis development is closely related to imbalances in lipid metabolism, including lipid uptake and/or lipid efflux inhibition (*Maguire, Pearce & Xiao, 2019*; *Poznyak et al., 2020*).

The purpose of "omics" research is to integrate and analyze a large amount of information that represents the overall biological parameters in a particular state through advanced computer technology to identify some important information related to phenotypes and/or diseases (*Vernon et al., 2019*). A single biomarker may not be sufficient to represent the complex biological processes of the disease, and omics methods can be used to capture multiple variables and demonstrate the interrelationship between those variables in the disease process (*Vernon et al., 2019*). In recent years, the expression profiles of atherosclerosis have been determined by omics approaches, such as microarrays and RNA-seq, and hundreds of differentially expressed genes (DEGs) involved in the development of atherosclerosis have been identified (*Liu et al., 2020*; *Tan et al., 2017*). Recently, many studies have attempted to explore the possible molecular mechanisms of macrophage-derived FCs (M-FCs) through microarray technology and bioinformatics analysis (*Hägg et al., 2008*; *Huang et al., 2019*; *Reschen et al., 2015*). However, few studies have systematically explored the molecular mechanism of foam cells derived from smooth muscle cells (SMC-FCs) through high-throughput methods. In this study, bioinformatics techniques were utilized to investigate the possible mechanisms of SMC- and macrophage-derived FCs to identify the common molecular mechanisms of these two different sources of FCs.

## METHODS AND MATERIALS

### Data resources

The GEO database (http://www.ncbi.nlm.nih.gov/geo) is an open public database established by the National Center for Bioinformatics (NCBI), containing a large amount of data from microarrays, gene chips, and RNA-seq (*Riksen & Stienstra, 2018*). To obtain the genes associated with the formation as well as the development of atherosclerosis, the dataset consisted of human atherosclerotic plaques and normal arteries or early and advanced atherosclerotic plaques that were searched in the GEO database, and two datasets (GSE28829 and GSE43292) were obtained. Then, the in-vitro dataset contained ox-LDL treated smooth muscle cells (SMC) or macrophages that were screened out to identify foam cell-related DEGs of the two cell types. Furthermore, because of the important role of foam cells in the development of vulnerable plaques (*Liu et al., 2017*), the dataset composed of unstable vs. stable or ruptured vs. unruptured atherosclerotic plaques were screened to explore the expression of the foam cell-related genes in vulnerable plaques. The datasets of animal samples and the in-vivo human datasets of serum or plasma were excluded in this study.

The GSE28829 (*Döring et al., 2012*) dataset contains 13 early carotid atherosclerotic plaque samples and 16 advanced atherosclerotic plaque samples. GSE43292 (*Ayari & Bricca, 2013*) contains 32 normal carotid artery samples and 32 corresponding atherosclerotic plaque samples. GSE68021 (*Damián-Zamacona et al., 2016*) contains the gene expression dataset of human vascular SMCs simulated with oxidized low-density lipoprotein (ox-LDL) for 0 h, 1 h, 5 h, and 24 h (three technical replicates of each group). GSE54666 (*Reschen et al., 2015*) is a dataset of in vitro macrophage experiments including 6 samples of untreated macrophages and six samples of macrophage-derived FCs stimulated with ox-LDL for 48 h. GSE9874 (*Hägg et al., 2008*) consists of the gene expression profiles of macrophages treated or untreated with ox-LDL from 15 healthy subjects and 15 atherosclerotic patients. GSE41571 (*Lee et al., 2013*) contains data on 11 macrophage-rich regions from five ruptured plaques and six stable plaques. GSE120521 (*Mahmoud et al., 2019*) consists of an RNA-seq profile of four stable and four unstable plaque samples. The analysis strategy is shown in Fig. 1.

### Identification of DEGs

The downloaded gene expression profiles and their matched platform files were loaded into R (version 3.6.1) software and converted into gene symbol expression profiles. The LIMMA package was used to identify differentially expressed genes (DEGs) between the two groups (*Ritchie et al., 2015*). DEGs with adjusted $p$ values $< 0.05$ (adjusted by the Benjamini–Hochberg method) were considered significant. Co-DEGs were found in the overlap of different datasets as determined by an online web tool (http://jvenn.toulouse.inra.fr/app/example.html).

For the GSE68021 dataset, the DEGs of each time point (ox-LDL treated for 1 h, 5 h, 24 h) compared with negative control were identified. Furthermore, the Short Time-series Expression Miner (STEM) (*Ernst & Bar-Joseph, 2006*) program was used to analyze the

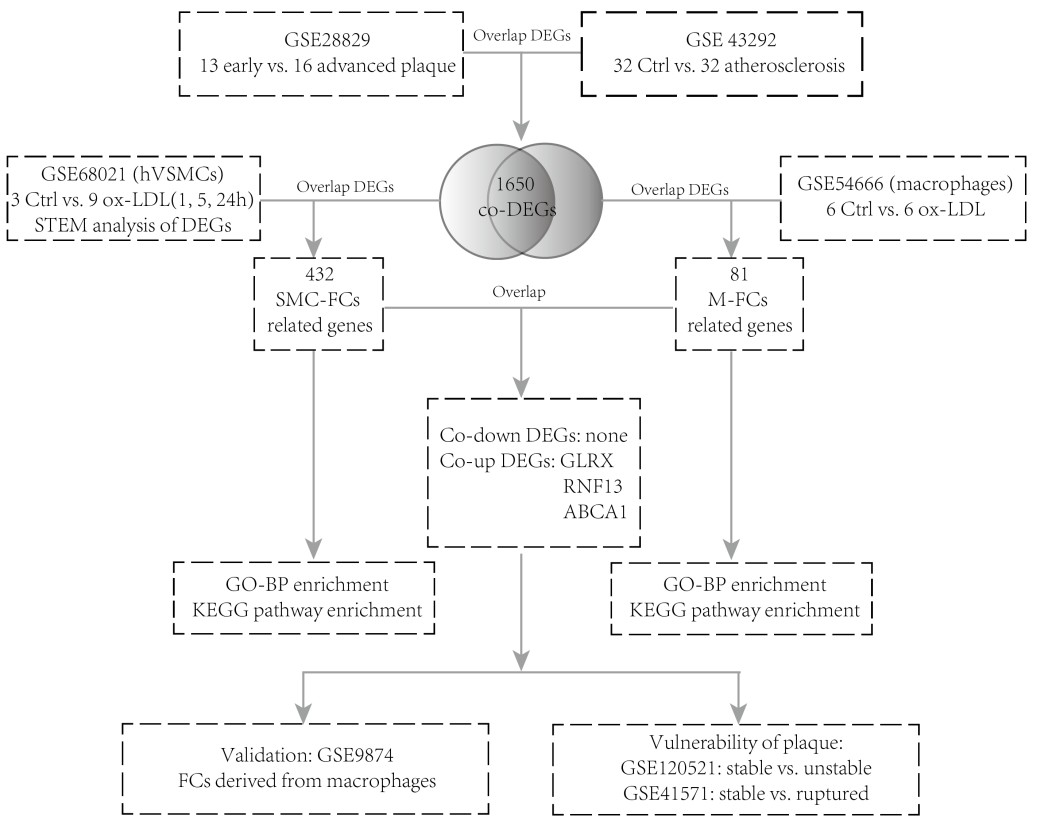

**Figure 1 Flow chart of the analysis.** DEGs, differentially expressed genes; NC, negative control; hVSMCs, human vascular smooth muscle cells; ox-LDL, oxidized low-density lipoprotein; SMC-FCs, smooth muscle cell-derived foam cells; M-FCs, macrophage-derived foam cells; PPI, protein-protein interaction; GO, Gene Ontology; BP, biological process; and KEGG, Kyoto Encyclopedia of Genes and Genomes.

temporal expression profiles of these DEGs. The significant clusters which manifested upregulated or downregulated characteristics over time were selected for the next analysis.

## Functional enrichment analyses

The purpose of gene function enrichment analysis is to determine the correlation between a group of genes and functional categories through a hypergeometric test. Gene Ontology (GO) is an international standardized gene functional classification system that covers three major categories: biological process (BP), cellular component (CC), and molecular function (MF). The Kyoto Encyclopedia of Genes and Genomes (KEGG) database is the major public pathway-related database for helping researchers understand the advanced functions of genes in biological systems. DAVID (https://david.ncifcrf.gov/home.jsp) is a very useful bioinformatics data resource for scientists to find meaningful biological information on genes or proteins (*Huang da, Sherman & Lempicki, 2009*). Herein, GO-BP and KEGG pathway analyses were performed using the DAVID database. The functional category with a *p*-value <0.05 (*Xia et al., 2019*) was considered significant.

## Validation of the common FCs-related genes

For the validation of the common FCs-related genes, the expressions of these genes were extracted out from the microarray dataset and analyzed by student's $t$-test. For the RNA-seq dataset, the normalized gene expression matrix of FPKM (fragments per kilobase of exon model per million reads mapped) values was downloaded and transformed into TPM (transcripts per kilobase million) values for comparison (*Li et al., 2010*). $P$ value <0.05 was defined as statistical significance.

# RESULTS

## Identification of DEGs associated with atherosclerosis

GSE28829 contained data on 13 early atherosclerotic plaques and 16 late atherosclerotic plaque samples. Through a LIMMA analysis, a total of 2,300 DEGs were identified in the advanced plaques, based on the gene expression in the early plaques (defined as atherosclerosis progression-related genes), including 1,150 upregulated genes and 1,150 downregulated genes (Fig. 2A). GSE43292 contained information on 32 nonatherosclerotic plaque samples (control) and 32 atherosclerotic plaque samples, and 6,378 DEGs were obtained, including 3,066 up- and 3,312 downregulated DEGs in atherosclerosis, based on the gene expression in the control group (Fig. 2B). After comparing the DEGs in the two datasets, 801 co-downregulated and 849 co-upregulated genes were identified (Figs. 2C and 2D). These co-regulated DEGs were defined as atherosclerosis-related genes.

## Identification of SMC-FCs-related genes and functional analysis

GSE68021 includes the gene expression data of human vascular SMCs treated with ox-LDL for 1 h, 5 h, and 24 h in vitro to simulate the status of SMC-FCs in atheroma plaques. In this study, a total of 3369, 4887, and 7170 DEGs were identified in 1 h, 5 h, and 24 h compared with control (Fig. 3A). Moreover, these DEGs obtained from different time points were investigated their expressive trends by using STEM program. As shown in Fig. 3B, 13 clusters with statistical significance were identified. And Cluster 1 (581 genes), 9 (601 genes), 12 (392 genes), 11 (413 genes), 26 (253 genes), 23 (240 genes) exhibited downregulated features over time, while Cluster 42 (630 genes), 48 (570 genes), 40 (493 genes), 29 (344 genes) exhibited upregulated trends over time. The expression of these genes with downregulated and upregulated trends were shown in Fig. 3C. Then these genes were compared with AS-related genes, and a total of 432 common genes were obtained and defined as SMC-FCs-related genes (Fig. 3D).

To clarify the biological functions of these SMC-FCs-related genes, the GO-BP and KEGG pathway enrichment analyses were performed, and a total of 71 BP and 25 KEGG pathway functions were enriched. As shown in Fig. 3E and Table S1, the enriched KEGG pathways were mainly involved in inflammation and contractive function pathways, such as Regulation of actin cytoskeleton (hsa04810), Vascular smooth muscle contraction (hsa04270), Chemokine signaling pathway (hsa04062), Leukocyte transendothelial migration (hsa04670). Similarly, the results of GO-BP terms suggested these genes were mainly involved in the contraction function, inflammation, cell cycle/apoptosis, and substance uptake and intracellular transport, such as actin filament organization
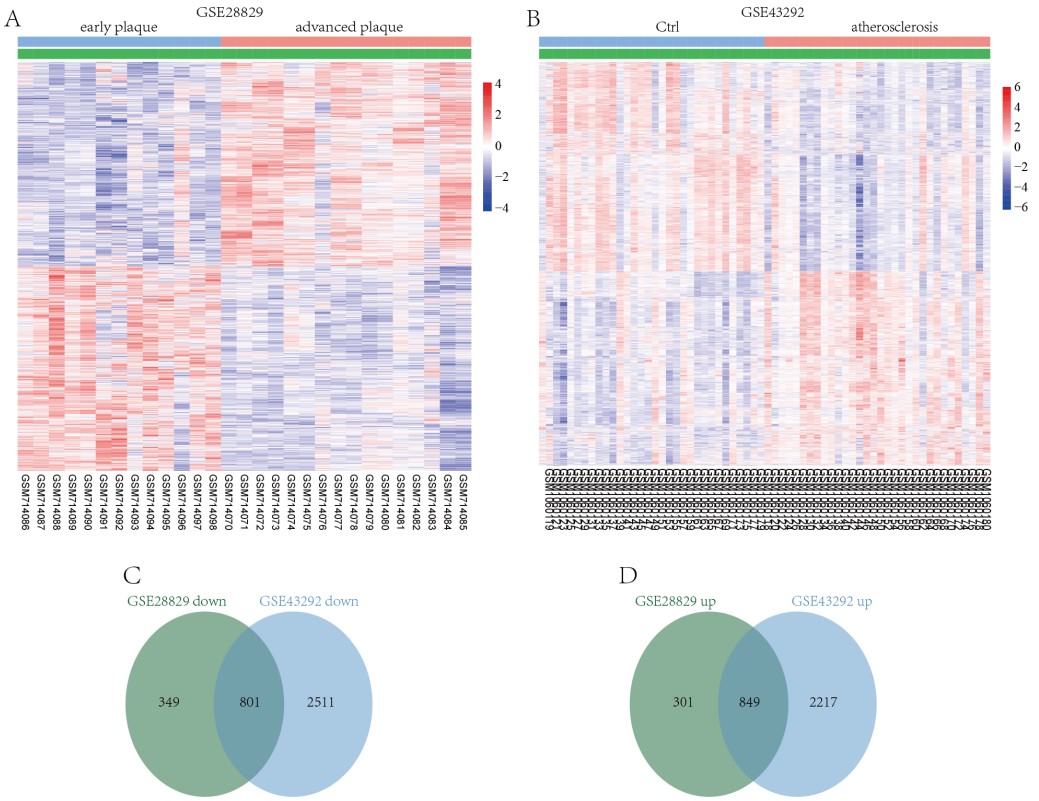

**Figure 2   Identification of DEGs associated with atherosclerosis.** Hierarchical cluster heatmap of DEGs in advanced atherosclerosis compared to those in early atherosclerosis samples in the GSE28829 dataset (A) and those in atherosclerosis samples compared to those in non-atherosclerosis samples in the GSE43292 dataset (B). Venn diagrams showing the co-upregulated DEGs (C) and co-downregulated DEGs (D) in the GSE28829 and GSE43292 datasets, which were defined as atherosclerosis-related genes.

(GO:0007015), smooth muscle contraction (GO:0006939), T cell receptor signaling pathway (GO:0050852), inflammatory response (GO:0006954), apoptotic process (GO:0006915), cell cycle (GO:0007049), early endosome to late endosome transport (GO:0045022), lysosome organization (GO:0007040) (Fig. 3F and Table S1).

## Identification of DEGs associated with macrophage-derived foam cells (M-FCs) and their functional enrichment

GSE54666 was the gene expression profiling of macrophages stimulated with ox-LDL in vitro to simulate the status of M-FCs in atheroma plaques. A total of 484 differentially expressed genes were obtained, including 259 upregulated genes and 225 downregulated genes (Fig. 4A). These DEGs and previously obtained AS-related genes were integrated, and the overlapping genes were defined as M-FC-related genes. As shown in Fig. 4B, a total of 81 M-FC-related genes were identified.

To explore the biological role of the 81 M-FCs-related genes, functional enrichment analysis was performed again using the DAVID database. A total of 30 GO-BP and 1 KEGG pathway were enriched. The GO-BP analysis mainly revealed lipid metabolism,

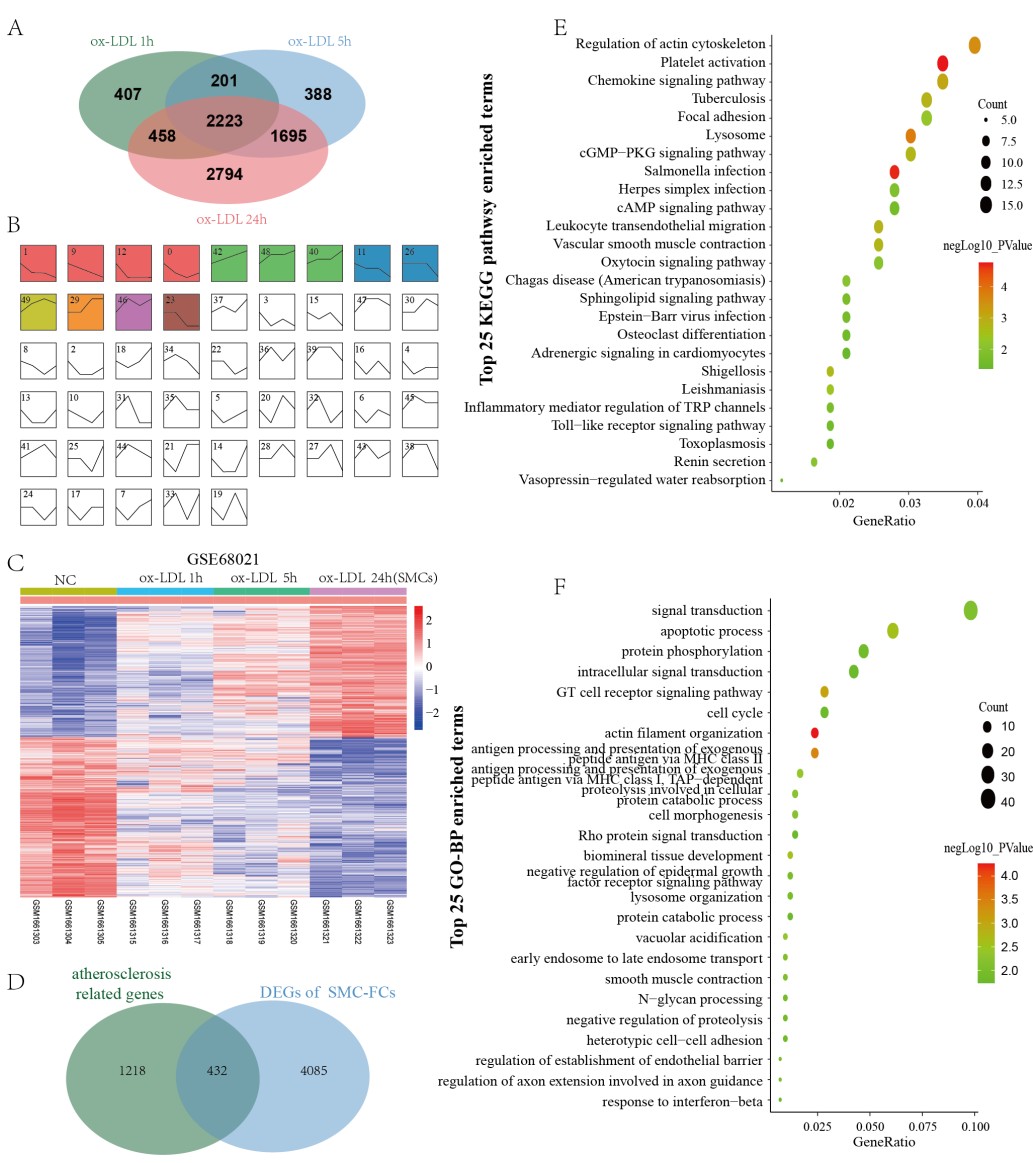

**Figure 3 Identification of SMC-FCs-related genes and functional analysis.** (A) Venn diagram showing the DEGs of each time point (ox-LDL treated for 1 h, 5 h, and 24 h) compared with control. (B) The expression trend clusters of all DEGs obtained above by STEM analysis. (C) Hierarchical cluster heatmap of DEGs with downregulated and upregulated trends overtime obtained by STEM analysis. (D) Venn diagram showing the overlap of DEGs in the ox-LDL-treated SMCs (with downregulated and upregulated trends overtime obtained by STEM analysis) and atherosclerosis-related genes, which were considered SMC-FCs-related DEGs. (E) and (F) The bubble diagram of the KEGG pathway and GO-BP enrichment analyses of SMC-FCs-related DEGs. In the bubble diagram, dot sizes represent counts of enriched DEGs, and dot colors represent negative $Log_{10}$ values ($p$ values). SMC-FCs, smooth muscle cell derived foam cells; GO, Gene Ontology; BP, biological process; and KEGG, Kyoto Encyclopedia of Genes and Genomes; ox-LDL, oxidized low-density lipoprotein.

foam cell differentiation, and immune response, such as negative regulation of cholesterol storage (GO:0010887), negative regulation of macrophage-derived foam cell differentiation (GO:0010745), response to low-density lipoprotein particle (GO:0055098), and negative

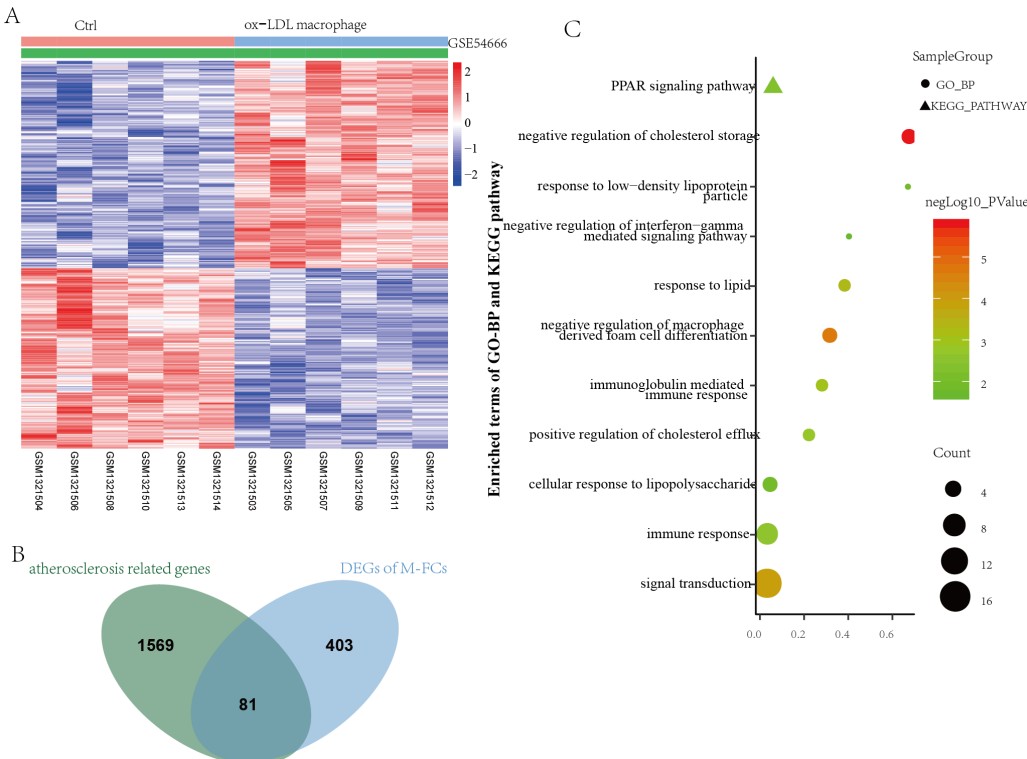

**Figure 4** **Identification of DEGs associated with M-FCs.** (A) Hierarchical cluster heatmap of the DEGs in the ox-LDL-stimulated macrophages compared to gene expression levels in the negative control from the GSE54666 dataset. Red represents upregulated genes, and green represents downregulated genes. (B) Venn diagram of the DEGs in the GSE54666 dataset compared to the AS-related genes and considered M-FC-related DEGs. (C) GO-BP and KEGG pathway enrichment analyses of the M-FC-related DEGs. In the bubble diagram, dot sizes represent counts of enriched DEGs, and dot colors represent negative Log$_{10}$ values (*p* values). DEGs: differentially expressed genes; M-FCs, macrophage-derived foam cells; ox-LDL, oxidized low-density lipoprotein; AS, atherosclerosis; GO, Gene Ontology; BP, biological process; and KEGG, Kyoto Encyclopedia of Genes and Genomes.

regulation of interferon-gamma-mediated signaling pathway (GO:0060336). The enriched KEGG pathway was the PPAR signaling pathway (hsa03320) (Fig. 4C and Table S2).

## The common genes of SMC-FCs and M-FCs related DEGs

To identify common molecular mechanisms in the SMC-FCs and M-FCs, the common genes were screened out. As shown in Fig. 5A, 15 common genes were identified: *CTSD, HHEX, TNFRSF21, GLRX, EDEM2, CTSC, LAT2, SPOCD1, ABCA1, LST1, CD74, PLAUR, BRI3, EMB,* and *RNF13*. Then, the expression trends of these 15 genes in AS tissue, SMC-FCs, and M-FCs were compared to that of the respective control. As shown in Figs. 5B–5C, all of these 15 genes were upregulated in the atherosclerotic group compared to the non-atherosclerotic group or in advanced plaques compared with early plaques. However, only four genes (*GLRX, ABCA1, HHEX,* and *RNF13*) exhibited upregulated trends in ox-LDL treated SMCs (Fig. 5D), while nine genes (*SPOCD1, PLAUR, CTSD, RNF13, ABCA1, BRI3, GLRX, TNFRSF21,* and *EDEM2*) were increased in ox-LDL treated
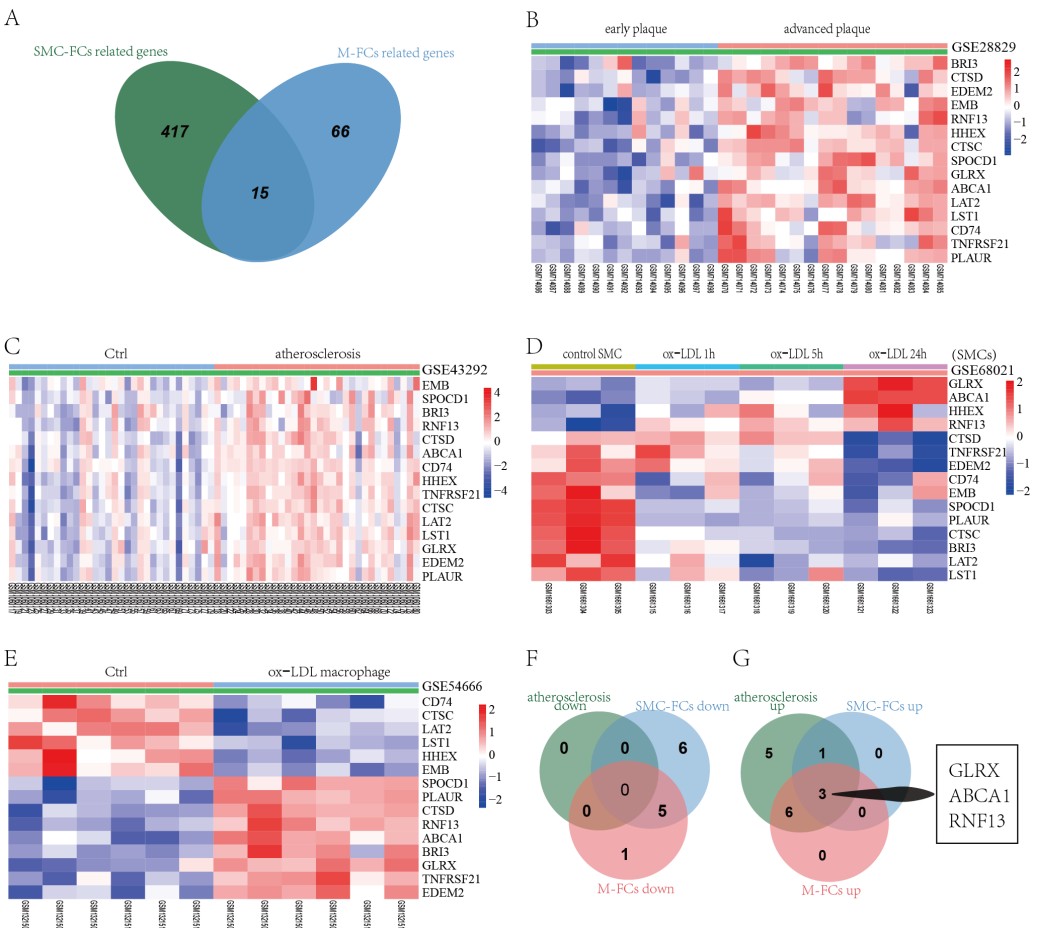

**Figure 5   Identification of co-regulated genes of SMC-FCs and M-FCs.** (A) Venn diagram of common DEGs related to SMC-FCs and M-FCs. (B–E) Cluster heatmaps of the expression of 15 common genes in the GSE28829, GSE43292, GSE68021, and GSE54666 datasets. Red represents upregulated genes, and green represents downregulated genes. (F) Venn diagram showing the co-downregulated genes and (G) co-upregulated genes in the atherosclerosis tissue, SMC-FCs, and M-FCs. SMC-FCs, smooth muscle cell-derived foam cells; M-FCs, macrophage-derived foam cells; and DEGs, differentially expressed genes.

macrophages (Fig. 5E). Then, the co-upregulated and co-downregulated genes were screened. The Venn diagrams showed the 3 (*GLRX, RNF13,* and *ABCA1*) co-upregulated genes; and no co-downregulated genes were found (Figs. 5F and 5G).

## Validation of the common FCs-related genes and their expression in vulnerable plaques

The expression of the three co-upregulated FC genes was then validated using another dataset. GSE9874 contains the gene expression profile of peripheral blood monocyte-derived macrophages treated or untreated with ox-LDL. As shown in Fig. 6A, all of the three genes were increased in the ox-LDL treated group (Foam cells group).

Previous studies have suggested that the formation and retention of the FCs could exacerbate atherosclerosis and fuel the development of the vulnerable plaques (*Bäck*

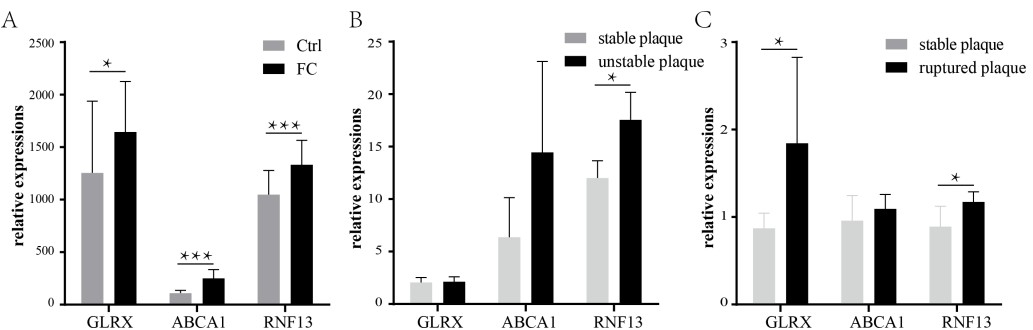

**Figure 6** **Validation of the common FCs-related genes and their expression levels in vulnerable atherosclerosis plaques.** Validation of the three co-upregulated genes: *GLRX*, *RNF13*, and *ABCA1*, in the macrophages treated with or without ox-LDL (GSE9874) (A). The expression of the three genes in unstable plaque samples (B) or ruptured plaque samples (C) compared to stable plaque samples in the GSE120521 or GSE41571 dataset, respectively. * $p < 0.05$, ** $p < 0.01$, *** $p < 0.001$. FC, foam cell.

*& Hansson, 2015*; *Liu et al., 2017*). Therefore, the expressions of the three FCs-related genes were investigated in the vulnerable plaques. As shown in Fig. 6B, only *RNF13* was statistically significantly upregulated in unstable plaques, while *ABCA1* and *GLRX* showed no statistical significance, though the means of *ABCA1* were larger in unstable plaques than stable plaques. In ruptured plaques, *GLRX* and *RNF13* were significantly upregulated compared with stable plaques, while *ABCA1* showed a slight increase with no statistical significance (Fig. 6C).

## DISCUSSION

Previous studies have suggested the crucial role of foam cells (FCs) in the formation and development of atherosclerosis, and the molecular mechanisms of FCs have been the focus of attention in recent years (*Chistiakov et al., 2017*; *Poznyak et al., 2020*). Recently, advanced high-throughput and bioinformatic technologies have been widely used for exploring the molecular mechanisms and predicting biomarkers for the diagnosis, treatment, and prognosis of the diseases (*Dona, Coffey & Figtree, 2016*). Through the gene expression microarray technology, Döring Y et al. revealed the upregulation of the plasmacytoid dendritic cell markers (such as *CD123, CD83, E2-2, CD317, CD32,* and *BDCA2*) in advanced versus early lesions (*Döring et al., 2012*). Ayari H et al. identified two genes (*CD163* and *HO-1*), which were significantly increased in atheroma plaques compared with intact arterial tissue by using GeneChip arrays (*Ayari & Bricca, 2013*). Lee K et al. identified 914 DEGs between stable and ruptured plaques by using microarray technology, and highlighted the involvement of FABP4 and leptin in the progression of atherosclerosis and plaque rupture (*Lee et al., 2013*). Mahmoud AD et al. revealed the DEGs between stable and unstable plaques, which were mainly linked with the plaque instability, including inflammation, matrix remodeling, and calcification (*Mahmoud et al., 2019*). Although these studies revealed some molecular mechanisms of the atherosclerosis development in various aspects, they could not reveal the cell-specific mechanisms, such as foam cells. Therefore, some studies tried to investigate the molecular changes of specific

cells by in-vitro experiments, such as macrophages and SMCs under the stimulation of ox-LDL (*Damián-Zamacona et al., 2016*; *Hägg et al., 2008*; *Reschen et al., 2015*). However, ox-LDL treatment alone might be much too oversimplified because of the complicated environment of cells in organisms. Therefore, in this study, we tried to combine the in-vitro and in-vivo experiments to relieve these influences.

In the present study, two datasets of gene expression in atherosclerosis tissues (GSE28829 and GSE43292) were analyzed, and 1650 co-regulated DEGs were identified. After screening these 1650 co-DEGs with DEGs of SMC-FCs (GSE68021) and M-FCs (GSE54666), a total of 432 SMC-FCs-related genes and 81 M-FCs-related genes were identified. To identify the common therapeutic targets of FCs, the common molecular mechanism between FCs derived from SMCs and macrophages was explored. Three co-upregulated genes were identified: *GLRX, RNF13,* and *ABCA1*. Furthermore, the expressions of these 3 genes in vulnerable atherosclerosis plaques were explored, and all of them were increasingly expressed in unstable or ruptured plaques compared to their expression levels in stable plaques, although some of the increased levels were not statistically significant.

Based on murine models, it was long presumed that all FCs in human atherosclerosis were derived from macrophages. However, subsequent studies demonstrated that resident cell types, especially SMCs, maybe the prominent originators of FCs (*Poznyak et al., 2020*; *Wang et al., 2019b*). However, macrophages are still the second most prolific sources of FCs and play crucial roles in the pathogenesis of atherosclerosis, in both the early and advanced stages (*Owsiany, Alencar & Owens, 2019*). Therefore, macrophages remain the most representative model of FCs and will continue to be studied.

The process of lipid metabolism in macrophages is divided into three main stages: lipid uptake, esterification, and efflux (*Maguire, Pearce & Xiao, 2019*). The lipid metabolism of macrophages in atheroma plaques gradually accelerates to be imbalanced, with excessive intracellular lipid deposition and leading to the formation of FCs (*Chistiakov et al., 2017*; *Maguire, Pearce & Xiao, 2019*). In the present study, a total of 81 M-FCs-related genes were identified, and their functions were found to be mainly enriched in lipid metabolism and immune responses.

The final stage of cholesterol metabolism is the efflux of cholesterol, which is mediated by transporters that depend on multiple key transcription factors, such as PPAR and LXLR (*Chistiakov, Bobryshev & Orekhov, 2016*; *Chistiakov et al., 2017*). These transporters mainly include ATP-binding cassette transporter 1 (ABCA1), ATP-binding cassette subfamily member G1 (ABCG1), and SR-B1, which play crucial roles in preventing the excessive intracellular accumulation of cholesterol and the formation of FCs (*DiMarco & Fernandez, 2015*). Similar to that of cholesterol internalization, a confusing relationship characterizes these efflux transporters during the progression of atherosclerosis. For example, a knockout of ABCA1 aggravated the formation of FCs but did not promote atherosclerosis plaque development (*Zhao et al., 2011*). Moreover, studies of ABCG1 and SR-B1 have also revealed controversial results, as they have a promoting role in early atherogenesis but a protecting role in advanced atherosclerosis (*Meurs et al., 2012*; *Van Eck et al., 2004*; *Zhang et al., 2003*).

SMCs also play crucial roles in all stages of atherosclerosis, and they are considered the leading sources of FCs (*Poznyak et al., 2020*; *Wang et al., 2019b*). During the process

of atherosclerosis, activated macrophages may secrete many inflammatory factors or cytokines that can promote the phenotypic transformation of SMCs from the resting systolic phenotype to the activated synthetic or "macrophage-like" phenotype. These phenotypic changes in SMCs are generally associated with the downregulated expression of contractile proteins and increased proliferation (*Zhang et al., 2012*). In the present study, a total of 432 SMC-FC-related genes were identified, and they were mainly involved in contraction function, inflammation, cell cycle/apoptosis, and substance uptake and intracellular transport, reflecting the roles of SMCs in atherogenesis. Moreover, compared with M-FCs, the expression of the cholesterol efflux transporter ABCA1 in SMC-FCs is much lower. Furthermore, the ABCA1 level in SMCs in advanced atherosclerosis was decreased compared to early atherosclerosis; however, no changes in macrophages were observed (*Allahverdian et al., 2014*). These results provide a plausible explanation for the high percentage of FCs derived from SMCs (*Allahverdian et al., 2014*).

RNF13 is an E3 ubiquitin ligase that is widely expressed in various animals and tissues. Previous studies have determined the roles of RNF13 in myogenesis, neuronal development, and tumorigenesis (*Zhang et al., 2009*; *Zhang et al., 2010*), in which it is mainly involved in cell proliferation (*Jin et al., 2011*). But few studies investigated the role of RNF13 in atherosclerosis or FCs. GLRX is a member of the glutaredoxin family, which highly contributes to the antioxidant defense system. Overexpression of GLRX could attenuate H2O2-induced apoptosis of endothelial cells (*Li et al., 2017*). However, GLRX inhibits endothelial cell angiogenic properties (*Matsui, Watanabe & Murdoch, 2017*). Though increasing evidence has suggested the important roles of oxidative stress in atherosclerosis (*Sharif et al., 2020*; *Zhang et al., 2020*), few studies have revealed the role of GLRX in atherosclerosis and the formation of FCs. Therefore, the roles of RNF13 and GLRX in the process of atherosclersis and FCs remain unknown, which needs further exploration.

The present study has several limitations. Firstly, the sample size of some datasets used in this study is small, which might result in some bias. For example, the GSE120521 contained only 4 stable and 4 unstable samples. Secondly, there may be some confounding factors, such as types and morphology of atherosclerotic plaques, age, gender, and drug treatment, as well as clinical complications, which might affect the interpretation of the results, and could not be identified in this study. Thirdly, interexperiment variability, such as different detection platforms and different resources of samples from different laboratories, may result in somewhat differences in gene expressions. Moreover, different cell types, like SMCs and macrophages, might show different changes of gene expression in ox-LDL treatment or pro-atheromatous environments in tissues. Therefore, only three co-up regulated genes were identified in this study between the two cell types of foam cells.

## CONCLUSION

In the present study, 432 SMC-FC-related genes and 81 M-FC-related genes were identified, and they were found to be mainly involved in lipid metabolism, inflammation, cell cycle/apoptosis. Furthermore, three co-regulated genes associated with FCs were identified: *GLRX*, *RNF13*, and *ABCA1*. Among these genes, the roles of *ABCA1* in the

atherosclerotic process have been widely described, but many contradictory results have been presented, suggesting that the roles of *ABCA1* in atherosclerosis need to be further explored. Moreover, the roles of *RNF13* and *GLRX* in atherogenesis remain unknown, and more experiments in the future are needed to confirm their roles. These three genes have a common expression tendency in both SMCs and macrophages, which may have implications for the development of possible targeted therapeutic drugs in the future.

### Funding

The present study was supported by the Guangdong Provincial Science and Technology Planning Project, Grant/Award, Number: 2017A070701013, 2017B090904034, 2017B030314109. There was no additional external funding received for this study. The funders had no role in study design, data collection and analysis, decision to publish, or preparation of the manuscript.

### Grant Disclosures

The following grant information was disclosed by the authors:
Guangdong Provincial Science and Technology Planning Project: 2017A070701013, 2017B090904034, 2017B030314109.

### Competing Interests

The authors declare there are no competing interests.

### Author Contributions

- Kai Zhang conceived and designed the experiments, performed the experiments, analyzed the data, prepared figures and/or tables, authored or reviewed drafts of the paper, and approved the final draft.
- Xianyu Qin conceived and designed the experiments, analyzed the data, prepared figures and/or tables, and approved the final draft.
- Xianwu Zhou analyzed the data, prepared figures and/or tables, and approved the final draft.
- Jianrong Zhou and Pengju Wen performed the experiments, prepared figures and/or tables, and approved the final draft.
- Shaoxian Chen performed the experiments, authored or reviewed drafts of the paper, and approved the final draft.
- Min Wu conceived and designed the experiments, performed the experiments, authored or reviewed drafts of the paper, and approved the final draft.
- Yueheng Wu and Jian Zhuang conceived and designed the experiments, authored or reviewed drafts of the paper, and approved the final draft.

### Data Availability

Data is available at NCBI GEO: GSE28829, GSE43292, GSE68021, GSE54666, GSE9874, GSE41571 and GSE120521.

## Supplemental Information

Supplemental information for this article can be found online at http://dx.doi.org/10.7717/peerj.10336#supplemental-information.

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
