# Peer review of "Analysis of genes and underlying mechanisms involved in foam cells formation and atherosclerosis development"

_PeerJ, doi:10.7717/peerj.10336_

## Round 0.1 · original submission · Major Revisions

The reviewers have raised many concerns about the study design (e.g. inclusion criteria, analysis strategy, and validation plan), and interpretation of the results. In addition, it's not clear what additional insights have been uncovered, compared to the original publications. I also have concerns about statistical methods. Specifically,

(1) The sample characteristics of the samples are not given. It's unknown whether there are confounding factors that need to be adjusted.
(2) The preprocessing of the expression data is completely missing.
(3) Using LIMMA for the time-course data GSE68021 and the RNA-seq data is not appropriate.
(4) The multiple testing correction method used is not stated.
(5) it's unclear if the results are robust to some variation in the analysis.

Although I grant a major revision, I cannot guarantee the final acceptance. In my view, some of the concerns are very substantial and are not be easy to address.

·

Basic reporting

The manuscript is overall easy to understand. The introduction is clear, focusing on foam cells in the context of atherosclerosis. Flow chart of the analysis helps to follow the different comparisons.
Major comments:
- The papers corresponding to the published datasets used in this paper have not been referenced and discussed. It would be important to explain how different this new analysis is compared to the ones done in the original papers.
- The discussion section does not correspond to a discussion of the results but instead to an extended introduction. For example, no mention of the results is done between lines 255 and 275. This needs to be modified and should include the following points. In particular, the authors should include a comparison to the previous analysis of the microarray/RNAseq and stating the novelty/difference of their approach. The detection of only a small overlap between SMC-FCs ad M-FCs should be discussed (due to cell type difference? experimental design?). The authors analysed data from in-vitro experiments as well as in-vivo/tissue. These analyses come with different limitations. Indeed, tissue heterogeneity will limit the interpretation of the result and this needs to be mentioned.
Minor comments:
- A few typos and unclear sentences can be found along the manuscript such as Line 82 “plague”; Line 206 “GSE5466 was expressed”; Line 244 “corelated”; Line 333 “compared its expressing in”
- Figures: Gene names in Heatmap should be removed if they are too small to be read. A title for the x-axis of the bubble diagram should be added and the size of the text should be increased.
- The manuscript contains many abbreviations that make the reading of the manuscript difficult. For clarity, some abbreviations could be removed (“atherosclerosis” shouldn’t be abbreviated to AS).
- In Figure 6B, all significant comparisons are shown while only a few comparisons are described in the text. It might be better to show only the statistics for the relevant comparison.

Experimental design

Overall, relevant questions and objectives are well defined with the flow chart helping to understand the analysis. Methods section is comprehensive.
Major comments:
- The choice of dataset to include in this analysis has not been explained clearly. Does it correspond to all publicly available data on foam cells and atherosclerosis? If only some dataset were included, can the authors justify their selection criteria.
- The choice and order of comparisons in this analysis are not completely claer for me. Indeed, the authors started by analysing 2 in-vivo dataset then 2 in-vitro datasets. Finally, they finish by analysing another in-vivo and in-vitro dataset, for validation. I do not understand why the 3 in-vivo datasets where not analysed and integrated together. What is the justification for choosing the dataset for target identification versus the dataset for validation?
Minor comment: The threshold (fold change and pvalue) to identify DEGs should be included in the Methods.

Validity of the findings

While the comparison of in-vitro and in-vivo dataset to identify genes involved in foam cell formation in atherosclerosis is interesting, the interpretation of results and discussion is sometimes missing.
Major comment: As mentioned above, the Discussion of the manuscript should include the limitations of this type of analysis

Additional comments

In summary, this manuscript provides an interesting analysis by integrating different datasets to find relevant candidates involved in foam cell formation. The different steps of the analysis and results are clearly explained. However, the choice of data comparison is not justified and does not seem pertinent. The authors only find 4 candidates, probably due to the limitations of this kind of approach. Discussion of the results and limitation should be incorporated to this manuscript.

Reviewer 2 ·

Basic reporting

Figures need to be remaking.

Experimental design

no comment

Validity of the findings

no comment

Additional comments

This manuscript provided a list of genes that are functional related to the development of atherosclerosis. The idea to utilize multiple existing datasets is valuable, though I am not fully convinced by the technical details of the proposed approach. My comments are list as follow.

Major comments:
1. For the DEGs associated with AS, the first gene sets come from DE analysis of early versus late atherosclerosis samples, and the second gene sets come from DE analysis of athrosclerotic versus nonathrosclerotic samples. It is not clear to me why you need to take the intersection of the intersection of the DEGs of the two comparisons. The first experiment design is comparing between patients with AS not health control. Theoretically, the second set of gene is enough. Please explain the specific purpose of including first gene sets.
2. To generate SMC-FCs related genes and M-FCs related genes. The authors simply take the overlap genes between studies. This approach ignores the difference of sample size across study designs and the effect size of each gene in DE analysis. It would be better to use a meta-analysis which adjusted for these effects.
3. To validate the four coregulated genes associated with FCs, the authors used vulnerable AS plaques dataset. It seems to be a logic gap need to be filled. It is not clear to me why the vulnerability of AS can validate the role of FCs in AS. Please explain it in more details.

Minor comment:
The gene names (row names) of Figure 2 and Figure 4’s heatmaps are unreadable. Please make a clearer figure or completely remove the gene name and put them in supplementary tables.

---

## Round 0.2 · Minor Revisions

Please address the reviewer's additional comments.

·

Basic reporting

no comment

Experimental design

no comment

Validity of the findings

no comment

Additional comments

My first review of the manuscript contained several major and minor comments. While a few of these comments have been addressed, I still have some issues.

The remaining comments are:

- While the authors justified the validation of their candidates in 2 additional in-vivo datasets looking at vulnerable plaque (in response to comment (9)), the authors do not explain why they choose GSE9874 to validate their candidates. I do not understand why this dataset was not analysed in combination with GSE54666. Can you justify this choice?
GSE9874 corresponds to -/+ oxLDL from 15 healthy/athero. The Figure 6 includes -/+ oxLDL conditions but there is no mention of healthy vs athero. Can you explain if both healthy and athero samples were used?

- The discussion still needs improvement. In comment (1) and (2), I mentioned that previous studies, that generated these datasets, have not been referenced and discussed. The revised discussion only comments on 2 out of the 7 studies. For this reason, I consider that my comment has not been fully addressed. To answer comments (3) and (8), the authors included a new section to the discussion corresponding to the limitation of these studies. The 2 new sections of the discussion (discussion of the datasets and limitations of the study) are vague and poorly written. The authors need to explain what they mean by “interexperiment variability”, “cell heterogeneity”. The authors should point out which datasets is affected by the limitation of the sample size. The “sequencing related technical errors” shouldn’t be mentioned within the “external clinical traits”.

Reviewer 2 ·

Basic reporting

no comment

Experimental design

no comment

Validity of the findings

no comment

Additional comments

The authors addressed all my comments in the revised version. I have no further questions or comments.

---

## Round 0.3 · accepted · Accept

Your manuscript is ready for publication. Congratulations!